# Impact of COVID-19 Pandemic on Hotel Employees in the Greater Accra Region of Ghana

Dolores Mensah Hervie [1,*], Ernest Amoako-Atta [2], Md Billal Hossain [1], Csaba Bálint Illés [3,*] and Anna Dunay [3]

[1] Szent István Campus, Doctoral School of Economic and Regional Sciences, Hungarian University of Agriculture and Life Sciences, Páter Károly utca 1, H-2100 Gödöllő, Hungary; hossain.md.billal@phd.uni-mate.hu

[2] Department of Landscape Planning and Regional Development, Faculty of Landscape Architecture and Urbanism, Hungarian University of Agriculture and Life Sciences, 1118 Budapest, Hungary; amoako-atta.ernest@phd.uni-mate.hu

[3] Szent István Campus, Institute of Agricultural and Food Economics, Hungarian University of Agriculture and Life Sciences, 2100 Gödöllő, Hungary; dunay.anna@uni-mate.hu

* Correspondence: dhervie@atu.edu.gh (D.M.H.); illes.balint.csaba@uni-mate.hu (C.B.I.)

**Abstract:** This study examines the effect of the COVID-19 pandemic on hotel employees in the Greater Accra Region of Ghana using the conservation of resource and human relations theories to ascertain the level of impact. Data was collected from 511 hotel employees from 58 hotels using questionnaires. The employees were randomly sampled. Stepwise Regression in Statistical Package for Social Sciences version 16 model was used to run the analysis. Nine independent variables were included in a stepwise regression model. Six came out as significant and explained 51.6% of the variation in the percentage of their salary that staff received during the current COVID-19 pandemic. The remaining 48.4% was explained by other factors such as the years of service and whether the facility shut down during the lockdown. It was further identified that about 80% of the respondents had their salaries reduced, and that work schedules and working hours were altered, particularly during the lockdown and closure of the country's borders. It is recommended that hotels should network, be more customer-oriented, be health- and safety-focused, frequently update their communication channels, and make digitalisation and human resource development a priority as measures to help the industry in its recovery process.

**Keywords:** Accra; COVID-19; hotel employees; lockdown; pandemic

## 1. Introduction

The COVID-19 epidemic has undoubtedly exposed the vulnerability of the hospitality and tourism industry to natural disasters and economic volatility. The continuous impact of the novel COVID-19 on the sector poses important questions about the rate of recovery of this industry. The working environment in the hospitality industry has been adversely affected, generating fears of the sector's ability to recover from the losses, turn around quickly, and employ skilled labour in the future [1].

After the World Health Organization's (WHO) declaration of the COVID-19 pandemic on 11 March 2020, over 90% of the global populace were confronted with massive restrictions due to lockdown of countries, ban on local and international travel, and shut down of airports and borders (policies to restrain the spread of the COVID-19) [2]. In addition, flights were cancelled, social gatherings were reduced to the barest minimum, sports leagues were cancelled, educational systems were closed, conferences were either postponed, held online, or cancelled, restaurants, bars, casinos were all closed down, and people were encouraged to stay home as part of the measures taken globally to curb the spread of the COVID-19.

Almost all industries in every economy across the globe have been affected by this turn of events. One of the most severely hit industries is hospitality and tourism, due to lockdowns, quarantines and travel restrictions. Globally, 62 million (18.5%) jobs were lost in this sector in 2020 compared to 2019 when the industry contributed 10.6% (334 million) of jobs worldwide. 80% of these job losses occurred among the small and medium enterprises in the hospitality and tourism sector, affecting mostly women, youth and minorities who form the majority of the workforce in the service industry [3]. According to Bajrami et al. [2], tourist destinations that were recognised to be predominantly over-utilised were deserted as a result of the restrictions and this plummeted the hospitality and tourism businesses affecting the workforce worldwide. In a study to investigate job market trends in the United States of America during the COVID-19 pandemic, [4], Huang et al. identified that low paying jobs in the hospitality industry were the worst affected and they projected that 24% of redundancies caused by COVID-19 may be perpetual. The study further suggests that the pandemic has brought to light concerns relating to occupational health and safety, knowledge gaps, technology implementation implications, and work restructuring in this sector. Already, studies have shown that employees in the hospitality and tourism sector are prone to work-related stress [5–7]. The COVID-19 pandemic has not only created socio-economic devastation of economies but also triggered emotional and mental trauma among employees due to persistent rises in job insecurity. Some employees in the industry were faced with exceptionally high rates of job uncertainty and its related mental trauma. In their contribution to this discussion, Khan et al. [8] examined the fundamental framework of this sector by analysing 372 hospitality industry employees during this COVID-19 pandemic. It was revealed that the threat of insecure employment facilitates the connection between economic crunch, joblessness, and psychological health. COVID-19 reinforces the link among economic crises, psychological health and seeming instability in employment.

The situation is not different in Africa as most governments responded rapidly with strict measures to control the spread of the virus. Taking Ghana as a typical example, on 15 March 2020, (four days after the World Health Organisation declared COVID-19 a pandemic), the government banned all school activities and social gatherings, followed by restrictions of movement of people in Accra, the capital city, and another major city, Kumasi. The country's borders and airports were further closed and both public and private organisations were encouraged to work from home. The outcome of the actions taken to reduce and eradicate the COVID-19 pandemic locally and internationally have had a direct toll on the hospitality industry and its workforce [9]. The yearly contribution from hotels and restaurants to Ghana's Gross Domestic Product (GDP) dropped from approximately USD 985.8 million in 2019 to USD 640.9 million in 2020, probably due to the COVID-19 pandemic [10]. The worsening financial conditions of hotels brought pressure on employment and job insecurity for the workforce in this sector in Ghana as also happened in other countries. Most hotels had to lay off some of their employees, salaries were reduced, some were asked to take voluntary leave, others had their routine duties and positions altered while some employees had their working hours reduced [11]. Ghana is a Sub-Saharan African country, that faces the Gulf of Guinea and the Atlantic Ocean to the south. It covers an area of 238,535 km$^2$ and shares borders with Togo in the east, Cote d'Ivoire in the west and Burkina Faso in the north. Ghana is endowed with unique natural, cultural and historic attractions which makes it one of the most preferred destinations in Sub-Saharan Africa. The Greater Accra Region houses the capital city of Ghana. This region is the smallest in size, but the second most heavily populated region in Ghana with about 2.7 million people. The region has many hotels due to its positioning and economic activities. It is also the seat of government [12,13].

The tourism and hospitality sector in Ghana has progressed over the years to become one of the fastest-growing sectors and is ranked as the fourth foreign exchange earner for the economy. This is due to significant foreign direct investments in the mining sector, which has led to the influx of foreigners into Ghana. Foreigners who travel into the country utilise hotels for their conferences, businesses, training, leisure and so on. Other factors

that have contributed to the sector's growth include the increasing middle-income class, and the political and economic stability that the country enjoys.

Ghana's labour market is typified by informal recruitment. The majority of the Ghanaian workforce have low qualifications and education, and this is reflected in the hospitality industry. According to XINHUA [14], Ghana's unemployment rate rose to a record high of 13.4% in 2021 from 6% in 2010 due to the repercussions of the COVID-19 pandemic. About 32.8% of Ghanaian youth from aged 15–24 are without jobs. The ILO [15] approximated that 275,000 tourism and hospitality jobs in Ghana were at risk due to the COVID-19 pandemic, while about 40% of hotel employees had taken their annual leave and a quarter had been laid off. Some of the affected employees found themselves back into the hotel industry after the restrictions, while others ended up in the informal sector or became entrepreneurs. The ILO [15] suggested that employment in the hotel industry may return to pre-pandemic status if people feel safe to travel and there is an easing in restrictions of movement across the globe. It is on this premise that the present study aims to assess the impact of the COVID-19 pandemic on hotel employees in the Greater Accra Region.

## 2. Reviewed Literature

The COVID-19 pandemic like any pandemic has taken its toll on economies across the globe and the hospitality sector is one of the hardest hit. This section details secondary information gathered on the subject.

### 2.1. Definition and Theory

Most COVID-19 infect animals, birds, and mammals. There are seven types of COVID-19 that are known to affect human beings. Four of them cause mild to moderate respiratory illness (HCoV-OC43, HcoV-HKU1 and HcoV-229E) while the remaining three cause acute and deadly respiratory illness (MERS-CoV, the Middle East Respiratory Syndrome (MERS) and SARS-CoV-2). The symptoms include cough, cold and fever [16]. A highly contagious strain B.1.1.617.2 of the SARS-CoV-2 known as the delta variant was first identified in December 2020. According to UNICEF [17], this highly transmissible delta variant was identified in India and spread to 142 countries (including Ghana) as of 10 August 2021, surging the number of total cases across the globe. For instance, the delta variant daily infection rate of 50 in June shot up to 100 in July and increased to 350 in August 2021. Another highly contagious variant of COVID-19 known as Omicron (B.1.1529), was first detected in South Africa in November 2021 and has since spread to many countries in the world. However, the availability of vaccines have minimised its catastrophic effect.

Brooks [18] defines a pandemic as "an epidemic occurring worldwide, or over a very wide area, crossing international boundaries and usually affecting a large number of people". [19] describes the concept of the pandemic as an "important global event that spans many centuries, which includes diseases of very different etiologies that exhibit a variety of epidemiologic features." According to Qiu et al. [20], the term 'pandemic' was derived from the Greek words 'pan' and 'demos' that mean 'all' and 'the people'. They reiterated that 'pandemic' suggests "a widespread epidemic of contagious disease throughout the whole of a country or one or more continents at the same time." They restated that the WHO standard definition of pandemic influenza means "a situation in which a new and highly pathogenic viral subtype, one to which no one (or few) in the human population has immunological resistance and which is easily transmissible between humans, establishes a foothold in the human population, at which point it rapidly spreads worldwide". From the aforementioned definitions, COVID-19 was declared as a pandemic.

The United Nations International Strategy for Disaster Reduction defines 'hazard' as "a dangerous phenomenon, substance, human activity or condition that may cause loss of life, injury or other health impacts, property damage, loss of livelihoods and services, social and economic disruption, or environmental damage" [21]. Gencer [22] describes disaster "as a sudden event, such as an accident or natural catastrophe that causes great damage or loss of life". UNISDR [21] again defines disaster as "a serious disruption of the

functioning of a community or a society causing widespread human, material, economic or environmental losses which exceed the ability of the affected community or society to cope using its own resources."

From the above definitions, the COVID-19 pandemic could also be termed as disastrous as well as hazardous.

The study of Shapoval et al. [1] used a combination of social systems theory and Hofstede's cultural dimensions to decipher the effect of COVID-19 on the hospitality industry in three countries: United States of America, Sweden and Israel. It focused on the opinions of selected managers in the hospitality industry, revealing that the effect of the COVID-19 epidemic has been experienced in all facets of life by individuals, societies, businesses, all sectors of economies and the world at large. Most of the respondents in the hospitality industry expressed despair, anxiety, apprehension, rejection, and were concerned about family wellbeing as well as a rise in job insecurity. They found that the restrictions on travelling had a substantial effect on the tourism and hospitality sector globally.

This study is driven by the conservation of resources and the human resource management theories to determine the impact of the COVID-19 pandemic on hotel employees in the Greater Accra Region. Holmgreen et al. [23] argued that the conservation of resources theory is about stress. Stress is an outcome of situations that threaten valued resources or cause their actual loss. Further, it is the quest to protect, preserve and obtain these cherished resources which motivates human behaviour when stressed. Human resource management theory postulates that the direction of human relations theories is about establishing and merging social, economic, and psychological goals of members and the organizational goals of production, productivity, and profit [24]. The reduction of salaries, working hours, and work schedule among others may cause stress on the respondents leading to a reduction in their productivity.

### 2.2. Impacts of COVID-19 on Hotel Employees

Seddighi [25,26] indicates that the COVID-19 pandemic is the worst crisis that has affected humanity since the Second World War. Its repercussion has been enormous, affecting men and women differently calling for more equality in gender-related policies. The UNTWO [27] showed that the pandemic has put an estimated amount of between 100 million and 120 million direct tourism jobs in danger, most of which are micro, small and medium enterprises (MSMEs).

In a study to examine the impact of COVID-19 on restaurants, using data from OpenTable platform (an online restaurant reservation company), sampling 50 restaurants from five advanced countries (Australia, USA, Canada, Germany, Ireland, UK) and Mexico, it was revealed that countries that went through a strict lockdown and banned dine-in restaurants plunged the majority of their restaurants into financial crisis and some consequently closed down. Others maintained their operations and provided delivery service [28]. It was also observed that family-owned, and other restaurants had to lay off more than 80% of their employees as a cost controlling measure. The Private Sector Job Quality Index (JOI—a system that evaluates job quality in the US by comparing desirable higher-wage/higher-hour jobs versus lower-wage/lower-hour jobs), expected an estimated number of 10.8 million employees working in bars and restaurants to lose their jobs due to the persistence of interruptions caused by the COVID-19 pandemic [28]. This study is consistent with the previous study conducted by Sogno [29], which indicated that key branded hotels embarked on redundancies due to the COVID-19 crisis, suggesting further that such hotel businesses in Europe and Asia were operating below 50% of their capacity. Marriott Hotel, for instance, operated below 75% of their normal level which led to the layoff of 4000 (two-thirds) corporate employees both at home and abroad.

Among the G20 countries where hospitality and tourism industry contribute 10% to employment, a six-month interruption led to 75% of job losses in the sector, which is equiv-

alent to above 7.5% decline in employment in an average G20 country, and the drop could exceed 10% in Germany, Italy, Mexico and Spain (well-known tourists' destinations) [30].

In another study to investigate employees' perspectives on the influence of the COVID-19 pandemic on their jobs and the Indonesian tourism industry as a whole, using a sample size of 52 employees from different tourism and hospitality companies, it was revealed that the hotels lost income as a result of low patronage due to the restrictions to limit the spread of the virus. Many employees who lost their jobs during this period were unskilled and low-income earners. Fifty percent (50%) of the respondents worked from home whilst some had their salaries reduced. Further to this, about 96.2% of the respondents revealed that they were individually affected by the pandemic [31]. This is supported by the study of Soehardi et al. [32], who tested a hypothesis to determine the level of significant impact of the COVID-19 pandemic on hotel employees in Jakarta. The results showed that there was a significant effect of the pandemic on the hotel employees (t value (2.221) > t table (1.99) with a $p$ value (0.027) < 0.05). The result was also consistent with data from the Tourism and Creative Economics, Jakarta Office that compared the total number of hotel staff in 2019 (31,282) to employment during the pandemic when the figure had declined by 59.9% (12, 518) [33].

The situation is no different in Africa. The World Bank [34] predicted that the economic downturn in Sub-Saharan Africa would plummet by −3.3%, the first recession in 25 years. An estimated total of 40 million people was driven into abject poverty in 2020 due to job losses, obliterating five years of progress in eradicating poverty. According to the IMF [30], travel and hospitality is extremely vulnerable to the COVID-19 pandemic because of its contact-intensive services. The sector may continue to struggle until people feel safe to travel again. Countries that depended heavily on tourism had the worst shock. For instance, the real GDP of such African countries (Mauritius, Seychelles etc.) and the Caribbean were expected to fall in 2020 by 12%, and Pacific Island nations like Fiji could plummet to 21%. A real-time study conducted in Senegal, Mali, and Burkina Faso implied that one out of four employees lost their jobs, and one out of two employees suffered a decrease in earnings as of the end of April 2020. It was also shown that employees in the hospitality and tourism industry, as well as SMEs, were immensely affected by the COVID-19 pandemic response measures because of the contact-intensive activities. The impact continued even after the lifting of the bans and lockdowns because of the lasting consequence of the economic downturn on employees [35,36]. The African Union was apprehensive that job losses in Africa could rise to 20 million as a result of the COVID-19 pandemic. In Ghana, approximately 10.5 million workers in the private sector (formal and informal) were susceptible to job cuts due to the pandemic, particularly employees in the hotels and restaurants, bars, education, entertainment and events, travel and tour operators in the hospitality and tourism industry, manufacturing, retail, and wholesale trade and MSMEs [37].

Like the countries mentioned earlier, the Ghana government took measures to limit the escalation of COVID-19, and the three-week (30 March–20 April 2020) partial lockdown had a significant direct negative impact on its hospitality and tourism industry. This is because the sector relies considerably on international visitors [34]. The sector lost about $171 million in revenue in 2020 which led to significant job losses, reduction in working hours and labour earnings. To ascertain the impact of COVID-19 on 3–5 Star hotels in Ghana, Danso et al. [38] revealed that before the COVID-19, majority of these hotels were operating within a range of 1–25%, however, it fell drastically during the pandemic.

The partial lockdown of the two most commercial cities in Ghana (Accra and Kumasi), the closure of Kotoka International Airport and the country's borders, as well as the limitation of social gatherings to 25 (later expanded to 100) brought economic activities to a halt. Flights were cancelled, funerals and other social events were postponed. In addition, special events like conferences, workshops, and trainings were either cancelled or postponed. People were encouraged to stay at home [39]. Companies had to resort to shift/rotational system for their employees. Some budget hotels had to shut down

completely due to lack of business, some large hotels closed down sections of their facilities leading to employee layoffs and salary reductions. Some employees were asked to stay home for some days in each month, others had their daily schedules altered whiles others had their working hours reduced. Most hotel employees suffered economic hardships due to the measures taken by their respective hotels. Although the bans have been lifted, employees in the sector continue to bear the brunt of the pandemic.

In order to continue to remunerate their employees, most hotels in Ghana that did not shut down during the period of the lockdown maintained a lean staff. [40]. This is in support of the findings of Dube et al. [28]. About 80% of informal small businesses in the hospitality sector also closed down and restaurants had an average decline in patronage of 60%. Furthermore, the Tour Operators Association recorded an estimated number of 11,558 cancellation of tourists' bookings, which led to a revenue loss of approximately USD 835,759.04 (GHC 4,847,402.41) [15]. Given a projected 5.5 million service employment in 2019, there was an estimated 0.8% negative growth, and this led to a possible 45,000 redundancies in 2020, in sectors including hospitality and tourism, transport and private education (sectors severely hit by COVID-19 pandemic) [37].

The Ghana Statistical Service in cooperation with the World Bank and the United Nations Development Programme also performed a COVID-19 Business Tracker Survey to determine the impact of the pandemic on businesses in Ghana, collecting data by interviewing 4311 firms between 26 May and 17 June 2020. The finding showed that approximately 770,000 workers (25.7% of the total labour force) had pay cuts, about 42,000 workers were laid off and almost 700,000 workers had their working hours reduced. The survey further revealed that sectors with highest closures even after lifting of the lockdown included education (63.0%), transport (34.0%) and hospitality (24.0%). The following are the details of the specific impact on the hospitality and tourism industry with regard to employment responses during March to June 2020. 6.7% of businesses in the hospitality industry laid off 5.0% of their workers, 22.6% of companies granted leave of absence to 19.5%, 23.1% of firms reduced the working hours of 23.2% of their workers and 33.8% of firms cut 30.5% of their employees' emoluments. These figures show an over 80% effect of COVID-19 pandemic on the sector [41].

Due the severity of the COVID-19 impacts on this industry, the National Hotel Association of Ghana (NHAG), in May 2020, petitioned the government among other things to suspend taxes and other levies paid by the sector for the period of the COVID-19 pandemic. They also requested for a 65% reduction on utility tariffs, and a cut on import duties for food and beverages. The association again solicited the government to allow retrenched employees of the sector to receive 50% of their Social Security and National Insurance Trust (SSNIT) [42]. Table 1 gives a summary of studies on the effects of COVID-19 on employees in the hospitality industry. It states the author (s), research method, the country of study and the key findings.

### 2.3. The Conceptual Framework

From Figure 1, in line with the objectives of this study, selected variables explaining the background of the respondents such as the sex, age, number of years served in the hotel and position held in the hotel were used as the background characteristics of the respondents. The selected intermediate variables which were considered to have direct effect on the dependent variable include variables asking for information as to whether the COVID-19 pandemic has affected the employee's work schedule, whether their respective hotels shut down parts of their facilities, the facilities that were shut down during the partial lockdown, as well and during the closure of the country's international airport and borders, and whether COVID-19 affected the staff working hours during the lockdown restriction imposed on the country. The variables constituting the background characteristics and the intermediate variables were treated in this study as the independent variables for the regression analysis. The dependent variable seeks to determine the percentage of salary received during the COVID-19 pandemic.

**Table 1.** Summary of studies on the effects of COVID-19 on employees in the hospitality industry.

| Author(s) | Method | Country(ies) | Findings |
|---|---|---|---|
| Abbas et al., (2021) [43] | The study used a time-lagged field survey to investigate the psycho-economic impact of job uncertainties among employees in the hospitality industry during the COVID-19 pandemic as well as the social support offered against the damaging consequences of the uncertainties. | Pakistan | The findings indicated that uncertainties in maintaining a job had a negative correlation with self-esteem and related positively with economic hardships. Additionally, assistance from society minimised the impact of job insecurity on self-respect, mental stability, financial independence and self-gratification considerably. However, where there was no support, the effects of work insecurity were severe, with economic hardships on workers of the hospitality industry. |
| Bajrami et al., (2021) [2] | The study sought to test how certain factors such as work insecurity, employee health, risk-taking and organisational changes affect occupational outlooks and employee attrition in the hospitality industry. Data was collected from 624 employees from the hospitality industry. | Serbia | It was discovered that organisational changes and uncertainties at the workplace were forecasters of negative results whereas risk-taking was a determinant of job satisfaction in the negative way. It was further revealed that age and marital status had a considerable effect on employee motivation and attrition. |
| Huang et al., (2020) [44] | The study used new high-frequency data on some small businesses in the US. Further, to decipher the states' policies and the daily confirmed cases of COVID-19 influence on labour markets, the study used mixed-effects regression models of state-level longitudinal data for two sectors (restaurants and recreation). | United States of America | The findings revealed that policies of business closure could be linked to economic and statistically considerable decreases in employment in the number of small businesses working in the hospitality sector. |
| Tu et al., (2021) [45] | To determine the extent to which COVID-19 influenced layoffs in the hospitality sector in China affected its employees' performance, the study collated field data from 302 employees and their supervisors during the two waves using in-role and extra-role performance. | China | It was specified that COVID-19-caused layoff survivors' COVID-19-related anxiety, resulting in decline in in-role and extra-role performance. |
| Yan et al., (2021) [46] | Data was collected from 211 hospitality employees in 76 hotels in Peru, through an online survey from 1 June 9 June 2020, to examine the level of their symptoms of depression suffered as a result of the impact of COVID-19 pandemic. Drawing on Transactional Theory of Stress Model and the employees' home and work environment. | Peru | It was shown that job satisfaction lessens the connection between COVID-19 risk sensitivity and employees in the hospitality industry as well as the probability of being depressed. However, the relationship is affected by family factors. |
| Mehta and Sharma, (2021) [47] | To assess the effect of COVID-19 and development of mindset on sustainable performance of hotels, the study used survey questionnaires and interviews to collate data from five-star hotels' employees, and analysed the data using qualitative approach. | India | The study discovered and confirmed the destructive effect of COVID-19 on the socio-economic sustainability of the hospitality industry. |

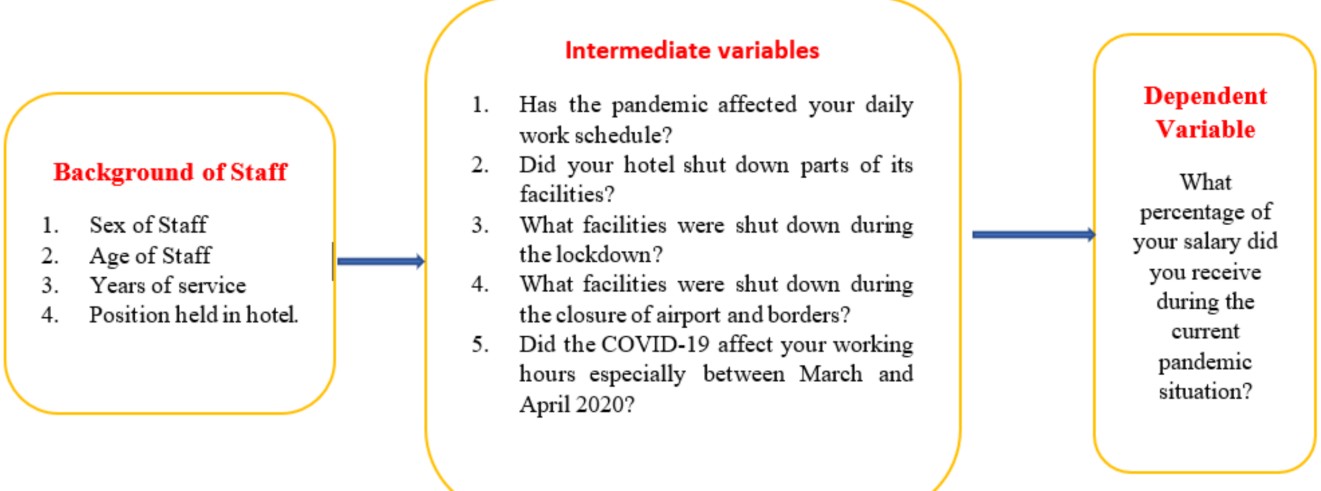

**Figure 1.** Conceptual framework of the effect of COVID-19 on hotel staff. Source: Authors' own construct, 2021.

## 3. Materials and Methods

This section describes research tools and methods used, and how they were conducted to meet the purpose of the study.

### 3.1. Data Collection and Analysis

Data was collected from 511 employees from randomly selected fifty-eight (58) hotels in the Greater Accra Region out of 813 registered hotels. This is because during the data collection (November and December 2020), most of the hotels were operating partially and others had shut down due to the impact from the pandemic. Open-ended questionnaires were distributed (physically) to a random sample of 601 hotel employees, based on 20% of the confirmed staff in each of the hotels. Five hundred and eleven (511) responded, representing 85% response rate. Statistical Package for Social Sciences (SPSS) version 16 was used to run the analysis of data. Nine independent variables were selected and were included in a Stepwise Regression model. Considering the nature of the dependent variable, multiple regression analysis was used with percentage of salary received during the COVID-19 pandemic situation as the dependent variable and other selected background characteristics of the respondents together with the intermediate variable used as independent variables for the regression analysis. Before the regression analysis, correlation analysis was conducted to determine the relationships as well as the direction of the relationship between the selected variables to be used for the regression analysis.

Figure 2 gives an illustration of the research approach. Primary data was collected from the selected hotels in the Greater Accra Region, which was captured and analysed with SPSS. Secondary information was gathered from the Ghana Tourism Board, World Travel & Tourism Council, the World Bank and scholarly articles.

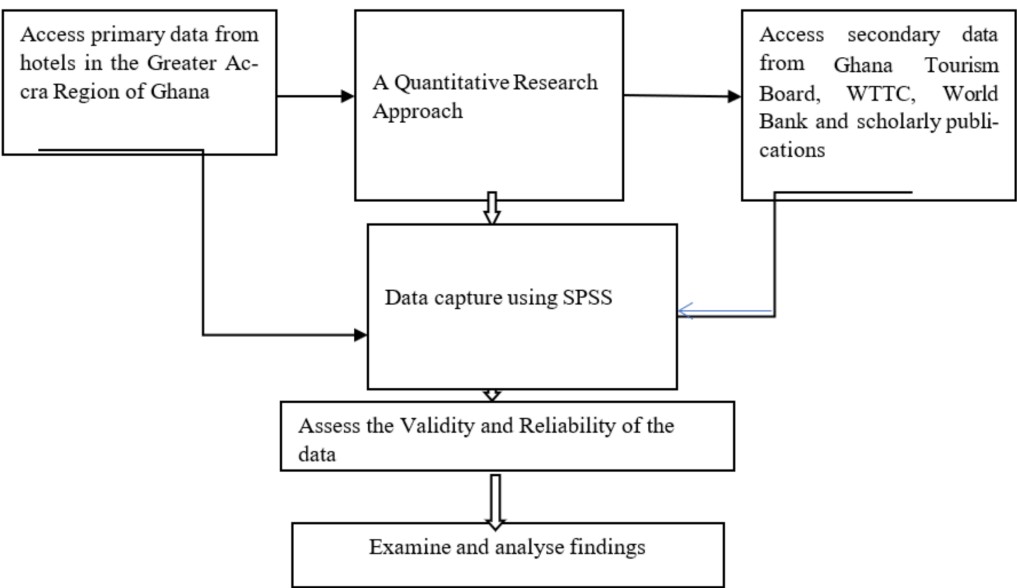

**Figure 2.** A representation of the Study Approach.

*3.2. Estimation Method*

Application of Stepwise Regression Model

Stepwise regression is a multiple regression analysis approach used to examine the relationship among a dependent variable and more independent variables. It blends the forward introduction and backward removal technique, considering the benefits of both methods. The assumption is that a selected variable becomes irrelevant when a new variable is introduced. The unimportant variable could be re-selected into the equation when it becomes necessary after the presentation of the new variable. Through this procedure, the realized regression equation maintains a few variables with the most significant effect, demonstrating the relationship between them efficiently and accurately [48]. Many studies have identified some key setbacks in the use of stepwise regression analysis such as prejudice when it comes to estimates and making confidence intervals extremely tight; there is an inherent discrepancy in the model selection processes (which is often ignored). The stepwise estimates are not invariant to insignificant linear changes [49]. In spite of these challenges, stepwise regressions remain a common tool for analysis, and are widely adopted among statistical packages. To overcome these challenges, among the variables selected, if there are two that are highly correlated, one of them was taken to avoid the effect of multicollinearity that might affect the reliability of the estimates of regression coefficients and also the value of the coefficient of determination (i.e., the proportion of the explained variance to the total variance represented by $R^2$ in the regression analysis [49].

The model equation could be written as:

$$Y = B_0 + B_1X_1 + B_2X_2 + B_nX_n + U \tag{1}$$

where: $B_0$ is a constant term, $B_1 + B_2 \ldots B_n$ represent the regression coefficients.

$X_1 + X_2$ where $X_n$ represent the independent variables, $Y$ represents the dependent variable while $U$ represents the residual terms (random error) [50].

**4. Results and Discussion**

The social and economic background characteristics of respondents (staff of the hotels) collated from the questionnaire indicate that male respondents were slightly more numerous (54.3%) than their female counterparts (45.7%). In terms of age, there were more younger respondents between the ages 20–40 years (68.5% of the total), followed by those aged 41–60 years (18.8%) while those above 60 years were about one-fifth (12.7%). Nearly 59% of the employees had dependents while 41.1% had no dependents. The majority of the

respondents (55.8%) had worked in the respective hotel for 1–5 years followed by those who had worked for 6–10 years (27.6%), while 16.6% had worked for 11–15 years. Nearly forty-two percent (41.9%) of the employees were staff without any position, 39.3% were supervisors and 18.8% were heads of department.

*Characteristics of Respondents and Effect of COVID-19*

Table 2 presents information on respondents and salaries received during the COVID-19 pandemic. It can be seen that a higher proportion of men (52.0%) compared to their female counterparts (31.3%) received between 76% to 100% of their salaries. More women (3.9%) than men (0.7%) received only 25% or less of their salaries during the COVID-19 pandemic period. A higher proportion (54%) of the younger staff of the hotels (aged 20–40 years) than their counterparts who were older (41+ years) had their full salary retained or reduced by 25%. In terms of position held in the hotel, it was revealed that a higher proportion of heads of departments in the hotels (77.1%) followed by the ordinary staff without any recognized positions in the hotel (47.2%) and lastly supervisors (20.9%) received between 76% to 100% of their salaries. The majority of the respondents (66.7%) who had worked for a relatively shorter period (1–5 years) in the hotels than their other counterparts (worked for 6+ years) had their salaries reduced by 25%. Surprisingly, respondents who could depend on relatives in the event that the COVID-19 pandemic extended for a long period (51.0%) received between 76% to 100% of their salaries compared to the lower percentage (38.8%) of their counterparts who indicated that they did not have anyone to rely on in the event that the COVID-19 extended for a very long time. It could be that most of the male employees had more work experience or higher education than their female counterparts. Further, the younger employees may have had a higher educational qualification, which could explain the higher salaries received.

**Table 2.** Background characteristics of respondents and percentage of salary received during the COVID-19 pandemic.

| Background Characteristics/Variable | | What Percentage of Your Monthly Salary Do You Receive? | | | | Total # of Resp. |
|---|---|---|---|---|---|---|
| | | $\leq$25% | 26–50% | 51–75% | 76–100% | |
| Sex | Male | 0.7 | 32.9 | 14.4 | 52.0 | 277 |
| | Female | 3.9 | 38.2 | 26.6 | 31.3 | 233 |
| Age of Respondent | 20–40 | 1.1 | 26.3 | 18.6 | 54.0 | 350 |
| | 41–60 | 5.2 | 64.6 | 10.4 | 19.8 | 96 |
| | 60+ | 3.1 | 40.0 | 43.1 | 13.8 | 65 |
| Position held in the Hotel | Staff | 1.9 | 43.0 | 7.9 | 47.2 | 214 |
| | Supervisor | 2.0 | 38.3 | 38.8 | 20.9 | 201 |
| | Head of Dept | 3.1 | 11.5 | 8.3 | 77.1 | 96 |
| Years of Service | 1–5 yrs | 0.7 | 21.4 | 11.2 | 66.7 | 285 |
| | 6–10 yrs | 1.4 | 63.1 | 17.7 | 17.7 | 141 |
| | 11–15 yrs | 8.2 | 35.3 | 54.1 | 2.4 | 85 |
| Do you have anybody to depend on in the event that you have financial difficulty over extended COVID-19? | Yes | 0.6 | 37.4 | 11.0 | 51.0 | 155 |
| | No | 2.8 | 34.3 | 24.2 | 38.8 | 356 |

The # sign in the table means 'number'.

Table 3 provides information on respondents and the effects of COVID-19 on their daily work schedule. It is clear that more males (52.5%) had their daily work schedule affected by the COVID-19 pandemic than their female counterparts (31.8%). Over sixty three percent (63.5%) of the respondents aged 41–60 years had their daily work schedules

affected, which was more than the other age groups. It is worth noting that a higher proportion of the hotel staff who did not occupy any position in the hotel (67.3%) had their daily work schedule affected by the incidence of the COVID-19 pandemic compared to the other staff who occupied positions such as supervisors (25.4%) and head of department (25.0%). In addition, a higher proportion (52.5%) of the respondents who had worked for 6–10 years had the effect of COVID-19 pandemic affecting their daily work schedules compared to the proportion of respondents who had worked in the hotels for relatively longer periods (11–15 years) or those who had worked less than 6 years, representing 28.2% and 42.5% respectively. Unfortunately, a higher proportion of the respondents who had nobody to depend on in the event of an extended effect of COVID-19 pandemic indicated that their daily work schedules were affected (46.9%) compared to the respondents who had someone to depend on (33.5%).

**Table 3.** Background characteristics of respondents and effect of COVID-19 on respondents' daily work schedule.

| Background Characteristics/Variable | | Has the COVID-19 Affected Your Daily Work Schedule? | | |
| --- | --- | --- | --- | --- |
| | | Yes | No | Total # of Resp. |
| Sex | Male | 52.5 | 47.7 | 277 |
| | Female | 31.8 | 68.2 | 233 |
| Age of Respondent | 20–40 | 37.7 | 62.3 | 350 |
| | 41–60 | 63.5 | 36.5 | 96 |
| | 60+ | 40.0 | 60.0 | 65 |
| Position held in the Hotel | Staff | 67.3 | 32.7 | 214 |
| | Supervisor | 25.4 | 74.6 | 201 |
| | Head of Dept | 25.0 | 75.0 | 96 |
| Years of Service | 1–5 yrs | 42.5 | 57.5 | 285 |
| | 6–10 yrs | 52.5 | 47.5 | 141 |
| | 11–15 yrs | 28.2 | 71.8 | 85 |
| Do you have anybody to depend on in the event that you have financial difficulty over extended COVID-19? | Yes | 33.5 | 66.5 | 155 |
| | No | 46.9 | 53.1 | 356 |

The # sign in the table means 'number'.

Analysis about how the COVID-19 pandemic has affected the work schedule of staff based on the background characteristics of the staff is presented in Table 4. From the table, whereas 56.7% of the male respondents indicated that they were affected by the COVID-19 pandemic only 25.8% of their female counterparts indicated that they were affected. The majority of the younger age group (20–40 years) as well as the older staff (60+ years) representing 52% and 50.8%, respectively, indicated that they were affected by the COVID-19 pandemic more than the middle age group (aged 41–46 years) where only 2.1% indicated that they were affected. Heads of departments and supervisors representing 46.9% and 45.8%, respectively, were affected by the COVID-19 pandemic more than their subordinate staff in the hotels (37.4%). As expected, the respondents who had worked or served in their hotels longer (11–15 years) had a lesser percentage indicating that they were affected by COVID-19 (3.5%) than their counterparts who had served relatively shorter period (62.4% for those who had worked for 6–10 years and 44.2% for those who had worked for 1–5 years). Unexpectedly, among the hotel staff who indicated that they had someone to depend on in the event that the financially difficult brought about by COVID-19 pandemic was extended, a lower percentage of them (29.7%) indicated that they were affected by the COVID-19 pandemic than their counterparts who indicated that they did not have anyone to depend on in the event that the financial difficulties brought about by COVID-19 were extended (48.0%). The results show that the COVID-19 pandemic affected the work schedule of most of the employees.

**Table 4.** Background characteristics of respondents and how the COVID-19 pandemic affected their work in general.

| Background Characteristics/Variable | | How Did COVID-19 Affect You and Your Work? | | |
|---|---|---|---|---|
| | | No Effect | Effect | Total # of Resp. |
| Sex | Male | 43.3 | 56.7 | 277 |
| | Female | 74.2 | 25.8 | 233 |
| Age of Respondent | 20–40 | 48.0 | 52.0 | 350 |
| | 41–60 | 97.9 | 2.1 | 96 |
| | 60+ | 49.2 | 50.8 | 65 |
| Position held in the Hotel | Staff | 62.6 | 37.4 | 214 |
| | Supervisor | 54.2 | 45.8 | 201 |
| | Head of Dept | 53.1 | 46.9 | 96 |
| Years of Service | 1–5 yrs | 55.8 | 44.2 | 285 |
| | 6–10 yrs | 37.6 | 62.4 | 141 |
| | 11–15 yrs | 96.5 | 3.5 | 85 |
| Do you have anybody to depend on in the event that you have financial difficulty over extended SARS-CoV19? | Yes | 70.3 | 29.7 | 155 |
| | No | 52.0 | 48.0 | 356 |

The # sign in the table means 'number'.

Information on the effect of COVID-19 on work hours between the period March and April 2020 of hotel staff were also analysed and findings are presented in Table 5. During this period (March and April 2020), a partial lockdown restriction was imposed on selected regions in Ghana including Greater Accra Region where the study was conducted. Exceptions were organization categorised as essential service providers such as the healthcare providers, selected hotels which were used as isolation centers, and security service providers among others. During that period, a lot of Ghanaians went through serious challenges and so in this study attempt have been made to elicit information from hotel staff as to how the lockdown affected their work schedule. From Table 5, it can be found that all the male hotel workers (100%) had their work schedule affected by the partial lockdown as against 92.7% of their female counterparts. Few of the younger hotel workers aged 20–40 years (constituting 4.9%) did not have their work schedule affected as compared to the older workers (41 years and above) where all of them (100.0%) had their work schedule affected. The data has again revealed that all the staff (100.0%) who do not occupy any position as well as heads of departments had their work schedules affected by the lockdown as against 91.5% supervisors. All the hotel staff (100.0%) who had worked for 1–5 years as well as those who had worked for 11–15 years had their work schedule (working hours) affected, as against 87.9% of those who had worked for 6–10 years. This implies that almost all the categories of employees were affected by the pandemic.

Correlation analysis was used as a means of providing further criteria to select variables for the regression analysis (see Table 6). In doing this, a thorough examination was made by correlating variables with each other. A look at the correlation matrix table shows that a majority of the variables had positive and high-level significant relationships (at the level of 99%) between variables. As indicated earlier, among the selected variables, one out of two highly correlated variables was dropped to avoid the effect of multicollinearity.

**Table 5.** Background characteristics of respondents and the effect of COVID-19 on work hours between period of lockdown (March and April 2020).

| Background Characteristics/Variable | | Did COVID-19 Affect Your Work Hrs. Especially between March and April 2020 (Partial Lockdown)? | | |
| --- | --- | --- | --- | --- |
| | | Yes | No | Total # of Resp. |
| Sex | Male | 100.0 | 0.0 | 277 |
| | Female | 92.7 | 7.3 | 233 |
| Age of Respondent | 20–40 | 95.1 | 4.9 | 350 |
| | 41–60 | 100.0 | 0.0 | 96 |
| | 60+ | 100.0 | 0.0 | 65 |
| Position held in the Hotel | Staff | 100.0 | 0.0 | 214 |
| | Supervisor | 91.5 | 8.5 | 201 |
| | Head of Dept | 100.0 | 0.0 | 96 |
| Years of Service | 1–5 yrs | 100.0 | 0.0 | 285 |
| | 6–10 yrs | 87.9 | 12.1 | 141 |
| | 11–15 yrs | 100.0 | 0.0 | 85 |
| Do you have anybody to depend on in the event that you have financial difficulty over extended COVID-19? | Yes | 89.0 | 11.0 | 155 |
| | No | 100.0 | 0.0 | 356 |

The # sign in the table means 'number'.

**Table 6.** Pearson correlation matrix.

| | Sex | Age | Years of Service | Position Held in Hotel | Q1 | Q2 | Q3 | Q4 | Q5 | Q6 |
| --- | --- | --- | --- | --- | --- | --- | --- | --- | --- | --- |
| Sex | 1 | 0.171 ** | 0.391 ** | 0.142 ** | 0.207 ** | −0.173 ** | −0.120 ** | −0.132 ** | −0.032 | 0.202 ** |
| Age | | 1 | 0.353 ** | 0.060 | 0.090 * | −0.303 ** | 0.160 ** | −0.131 ** | 0.017 | −0.116 ** |
| Years of Service | | | 1 | −0.008 | 0.059 | −0.449 ** | 0.020 | −0.308 ** | 0.349 ** | 0.096 * |
| Position | | | | 1 | 0.369 ** | 0.168 ** | −0.063 | −0.382 * | 0.054 | 0.058 |
| Q1 | | | | | 1 | 0.053 | −0.101 * | −0.045 | −0.134 ** | 0.161 ** |
| Q2 | | | | | | 1 | −0.103 * | −0.189 ** | −0.152 ** | −0.206 ** |
| Q3 | | | | | | | 1 | 0.149 ** | −0.255 ** | 0.348 ** |
| Q4 | | | | | | | | 1 | −0.426 ** | −0.213 ** |
| Q5 | | | | | | | | | 1 | −0.242 ** |
| Q6 | | | | | | | | | | 1 |

Source: Field Survey, 2020; ** Correlation is Sig. at the 0.01 Level (2 tail); * Correlation is Sig. at the 0.05 Level (2 tail).

After reviewing the empirical literature and considering Ghana's context, nine independent variables were deemed appropriate and were included in the stepwise regression model as follows: Sex of employee, Age of employee, Years of service in the hotel, Position held in the hotel, Has the pandemic affected your daily work schedule?, Did your hotel shut down parts of its facilities?, What facilities were shut down during the lockdown?, What facilities were shut down during the closure of airport and borders? and Did the COVID-19 affect your working hours especially between March and April, 2020? The dependent variable was: What percentage of your salary do you receive during the current situation?

Out of the nine variables, six (6) came out as being significant and explained 51.6% of the variation in the change percentage of salary that staff received during the current COVID-19 pandemic whereas the remaining of 48.4% was explained by other factors (see Table 7). From the table, among the 6 variables, the coefficient of determination of the standardised Beta showed that years of service in the hotel appeared to be the most important determinant (Beta = −0.357) to the change in percentage of salary that staff received during the current COVID-19 situation and accounted for 20.3% (i.e., $R^2$) of the total variation (reduction) in the percentage of salary staff received during the current COVID-19 situation. This means that the COVID-19 pandemic affected the salaries of the

workforce in the hospitality industry; more than 50% had their salaries reduced from 25% to 75%, particularly during the lockdown and closure of the airport. This finding is consistent with [38,39]. In both studies it was identified that employees' salaries and working hours were affected during the lockdown and airport closures.

**Table 7.** Result from the Stepwise Regression Model.

| Selected Independent Variable | Coefficients | | | t-Test | Model Summary | | | |
|---|---|---|---|---|---|---|---|---|
| | Unstandardised B | Std. Error | Standardised Beta | | R | $R^2$ | Change in $R^2$ | SE of the Estimates |
| Constant | 8.049 | 0.270 | | 29.857 | | | | |
| Years of service | −0.441 | 0.046 | −0.357 | −9.611 | 0.451 | 0.203 | 0.203 | 0.83124 |
| Facilities that were shut down during the lockdown | −0.240 | 0.015 | −0.617 | −16.019 | 0.570 | 0.325 | 0.122 | 0.76597 |
| Did the COVID-19 affect your working hours especially between march and April 2020? | −2.433 | 0.200 | −0.470 | −12.139 | 0.616 | 0.380 | 0.055 | 0.73491 |
| Facilities that were shut down during the closure of the airport and borders | −0.170 | 0.018 | −0.366 | −9.208 | 0.660 | 0.436 | 0.056 | 0.70174 |
| Age of respondent | −0.424 | 0.047 | −0.323 | −9.039 | 0.710 | 0.505 | 0.069 | 0.65794 |
| Did your hotel shut down parts of its facilities? | 0.266 | 0.079 | 0.119 | 3.349 | 0.718 | 0.516 | 0.011 | 0.65137 |

Source: Calculated from Field Survey, 2020.

The variables asking for facilities that were shut down during the lockdown, with a standardised Beta of −0.617, accounted for 12.2% of the percentage of salary that staff received during the current COVID-19 situation. Nearly 80% of the respondents indicated that parts of their hotel facilities were shut down during the lockdown and airport closure in Ghana. These facilities include swimming pools, drinking bars and guest rooms. The third important variable was a question asking for the COVID-19 effect on staff working hours, especially between March and April 2020 with a standardised Beta of −0.470 and accounting for 5.5% of the variations in the percentage of salary that staff received during the current COVID-19 situation. The 4th, 5th and 6th positions accounted for the following changes in the variation in the percentage of salary that staff received during the current COVID-19 situation, namely "Facilities that were shut down during the closure of the airport and borders, Age of respondent and Did your hotel shut down parts of its facilities", accounting for 5.6%, 6.9 and 1.1%, respectively, in the variation of percentage of salary that staff received during the current COVID-19 situation.

Dependent Variable: What percentage of your salary do you receive during the current pandemic situation.

Analysis of variance (ANOVA) Table 8 was used to test the multiple regression of the independent variables on the dependent variable. The purpose is to analyse the components of the total sum of squares and also test to determine if the inclusion of the independent variables was significant or not. As shown in Table 8, calculated F (6, 503, 0.01) = 89.227 which means that the independent variables were different from zero and hence the stepwise multiple regression of the independent variables on the dependent variable is highly significant. This result is commensurate with prior findings indicated in Table 1. For instance, Huang et al. [44], identified that the closure of businesses had a considerable effect on employment, particularly for employees working with SMEs in the hospitality sector. The study of Mehta and Sharma [47] also confirmed the destructive effect of COVID-19 on the socio-economic sustainability of the hospitality industry. Tu et al. [45] specified that

COVID-19-caused layoff raises survivors' COVID-19-related anxiety, resulting in decline in in-role and extra-role performance.

**Table 8.** Anova.

| Model | Sum of Squares | df | Mean Square | F | Sig. |
|---|---|---|---|---|---|
| Regression | 227.145 | 6 | 37.857 | 89.227 | 0 |
| Residual | 213.414 | 503 | 0.424 | | |
| Total | 440.559 | 509 | | | |

Source: Calculated from Field Survey, 2020.

Dependent Variable: What percentage of your salary do you receive during the current pandemic situation? As indicated earlier, more than 50% of hotel employees in Ghana had their salaries slashed between 25% to 75%, during the lockdown and airport closure.

The results of the analysis indicate that the COVID-19 pandemic has had a negative impact on employees in the hotel industry in the Greater Accra Region particularly on their salaries, during the partial lockdown and border closures, as supported by previous studies. Findings show that employees in the hospitality and tourism industry were faced with employment insecurity, anxiety, work related stress, mental health issues, COVID-19-caused layoffs and economic hardships. Furthermore, age and marital status had a considerable effect on employee motivation and attrition. Governments' policies to curb the spread of the COVID-19, which led to closure of businesses in the tourism and hospitality sector, also affected the human resources in the sector [43–46].

## 5. Conclusions and Recommendations

In conclusion, the COVID-19 pandemic has emphasised the extreme susceptibility of the hospitality and the tourism sector worldwide to natural disasters and epidemics. It is clear that the industry must draw up a stringent sustainable recovery plan to assist in navigating through the pandemic and beyond.

This is consistent with what happened in many countries across the globe including Africa. Some employees were asked to stay at home for a certain number of days in the month, a few of the staff were themselves victims of the COVID-19 virus whereas others had some of their family members infected by the virus. All of the above may have economic implications for hotel employees. Support systems such as physical and health programmes should be introduced to help employees recover from the pandemic shocks. As suggested by Kim et al. [51], hotels should continuously enhance their employees' productivity by investing in high-performance work systems which would eventually help in the recovery and growth of the hotels leading to a thriving industry. Zhang et al. [52] state that during the COVID-19 pandemic, hotel clients search for information about hotels directly before they make their preferred choices to minimise their physical risks. It was recommended that hotels should regularly update their information through effective communication channels such as the social, electronic and print media. They further identified that adhering to hotel workers' advice was the main risk-lessoning strategy for hotel clients in this period of COVID-19 pandemic. Information given by hotels and their staff are paramount to their clients; hence, the hospitality industry must provide relevant official information to their customers through the most appropriate channels. As suggested by Kovács et al. [53] and Reményik et al. [54] sectoral collaboration should be improved to enhance planning and policy management, particularly for policies relating to human resource development and technology.

Again, measures such as pandemic shock, savings, insurance for employees against global pandemics, and access to pension funds in the wake of a pandemic, should be implemented. Finally, hotels should build the capacity of their employees, regularly training them and developing their competencies. It is also recommended that further studies could be carried out to ascertain the level of impact of the COVID-19 pandemic on

male and female employees, respectively, as well as steps the industry could take to reduce its susceptibility to natural disasters and epidemics.

*Limitation*

Some hotels were partially operating, and others had shut down, particularly, budget hotels. Hence, they were not in a position to administer the questionnaires.

Some were skeptical and reluctant to divulge the necessary information. However, the researcher assured them of anonymity and confidentiality of the entire exercise.

**Author Contributions:** Conceptualisation, D.M.H., C.B.I., A.D.; methodology, D.M.H., C.B.I.; resources, D.M.H., E.A.-A. and M.B.H.; data curation, D.M.H.; writing—original draft preparation, D.M.H.; writing—review and editing, E.A.-A., M.B.H., C.B.I. and A.D.; visualization, D.M.H., E.A.-A., A.D. and M.B.H.; supervision, C.B.I. and A.D. All authors have read and agreed to the published version of the manuscript.

**Funding:** This study received no external funding.

**Institutional Review Board Statement:** Not applicable.

**Informed Consent Statement:** Not applicable.

**Data Availability Statement:** The data used in this study are available on request, for the reason of confidentiality.

**Conflicts of Interest:** The authors declare no conflict of interest.

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
