# Peer review of "Impact of COVID-19 Pandemic on Hotel Employees in the Greater Accra Region of Ghana"

_sustainability, doi:10.3390/su14052509_

Round 1

Reviewer 1 Report

The topic addressed in the article is very important and up-to-date.  

The abstract lacks a clearly stated objective of the study. In the introduction, the objective is stated as to assess the impact of the COVID-19 pandemic on hotel workers in the Greater Accra region. What exactly does this mean? Working conditions, financial situation, behaviour, etc.? In my opinion, the objective is very broadly outlined, especially since it is not further specified in the form of research problems. In my opinion, the article should be supplemented with research problems, which should be defined on the basis of the literature. The reviewed literature presents the latest scientific debate on the topic, but it mainly concerns statistical analysis and citation of other authors' studies.

In the article the authors instead of citing the names of the authors of the articles or studies use references e.g. [28] predicted that ... In my opinion this is not appropriate

Research methods were selected correctly. Conclusions and recommendations are very general, need refinement.

Author Response

Dear Sir/Madam,

Thank you.

Reviewer 2 Report

Comments and suggestions for authors are in attachment. 

Author Response

Dear Sir/Madam

Thank you.

Reviewer 3 Report

Researching the impact of the COVID-19 pandemic on various aspects of the tourism market is currently very fashionable, up-to-date and necessary. The problems faced by the tourism industry during the COVID-19 pandemic have a significant impact on many people's lives, mainly tourism workers and industries cooperating with tourism companies. In this respect, the article should be considered very topical.

This article omits the description of the scale of tourism in Ghana and the Greater Accra region. What is the structure of the tourism market? Number of hotels, number of tourists, number of employees in the hospitality industry. The reader should know the scale of the problem. This part of the article should be improved.

The second significant problem is the description of the labour market in Ghana. What is the scale of unemployment, and is it easy to find a job in other sectors of the economy? Were employees able to find another job when they did not have the opportunity to work in hotels? In Europe, there is a problem of reluctance to return to work in the hotel industry. Hotel workers who found other jobs outside the hospitality industry during the COVID-19 pandemic do not want to return to work in hotels, restaurants and other tourism businesses.

The downside is that the model built by the authors only includes employees who are still working in hotels. It does not include those who have left the hospitality industry because they have not agreed to wage cuts, working hours and other restrictions. This problem requires clarification as the authors do not seem to notice it.

The methods of statistical analysis should be considered standard and correct.

Author Response

Dear Sir/Madam

Thank you.

Round 2

Reviewer 2 Report

Thanks to the authors for the modified version of the manuscript.